# Deep Multi-Head CNN for Unified Breast Density Segmentation and BI-RADS Estimation in 2D Mammograms

**Obinna Agbodike**[*1] (iD)                              OBINNADYKE@GMAIL.COM
**Chang-Fu Kuo**[1,2] (iD)                                ZANDIS@GMAIL.COM
**Jenhui Chen**[1,3] (iD)                            JHCHEN@MAIL.CGU.EDU.TW

[1] *Center for Artificial Intelligence in Medicine, Chang Gung Memorial Hospital, Taoyuan, Taiwan*
[2] *Division of Rheumatology, Allergy and Immunology, Linkou Chang Gung Memorial Hospital, Taoyuan, Taiwan*
[3] *Department of Computer Science and Information Engineering, Chang Gung University, Taoyuan, Taiwan*

**Editors:** Accepted for publication at MIDL 2025

## Abstract

Comprehensive breast density estimation is crucial in mammogram assessment and cancer risk stratification, yet many existing AI-based radiomics methods designed for this purpose often tackle tissue segmentation and classification as separate tasks. To address this limitation, we propose a multi-head convolutional neural network (MH-CNN) that integrates these functions into a unified end-to-end architecture. Built on a ResNet101 encoder, our approach learns high-level features for breast density segmentation while parallel network heads perform continuous density regression and BI-RADS classification overlaid in the resulting images. Evaluation on the VinDr mammogram dataset yielded a Dice coefficient of 84.57% for segmentation, a mean absolute error (MAE) of 5.92% for density regression, and 80.51% accuracy for BI-RADS classification. These results suggest that the MH-CNN can streamline clinical workflows by providing objective and reliable breast density assessments.

**Keywords:** Breast cancer, CNN, Mammography, Radiomics, Segmentation.

## 1. Introduction

Breast density is a key imaging biomarker in mammography and a significant independent risk factor for female breast cancer (Gudhe et al., 2022). As part of its standardized evaluation framework, the American College of Radiology has established the Breast Imaging-Reporting and Data System (BI-RADS), which classifies breast density into four categories ranging from 0–100% and distinguished by a 25% increment. High percentage breast density (PBD), particularly evident in BI-RADS categories 3 and 4, is associated with an increased risk of cancer and reduced mammogram screening sensitivity due to the potential obscuration of tumors by radiopaque fibroglandular tissues (Behravan et al., 2024). Given that breast cancer remains one of the most frequently diagnosed malignancies worldwide and the leading cause of cancer-related mortality among women (Chen et al., 2021) thus emphasizes the critical need for comprehensive mammogram interpretation strategies. In this context,

---

* Corresponding author

accurate and automatic radiomics-based assessment of breast density is essential, not only for aiding radiologists and junior doctors in risk stratification but also for addressing the inherent subjectivity and inter-observer discrepancies that often characterize typical human visual analysis. However, current deep learning approaches for PBD assessment predominantly focus solely on either segmentation (Larroza and Llobet, 2022) or classification (Wu et al., 2017), or treat both tasks as separate processes (Saffari et al., 2020).

## 2. Method

We introduce a multi-head convolutional neural network (MH-CNN) that integrates segmentation, regression, and classification tasks into a unified end-to-end framework. Leveraging a shared ResNet101-backbone encoder for feature extraction, our MH-CNN performs breast density segmentation, continuous density regression, and BI-RADS classification concurrently via dedicated parallel heads. The model's network layout is depicted in Fig. 1.

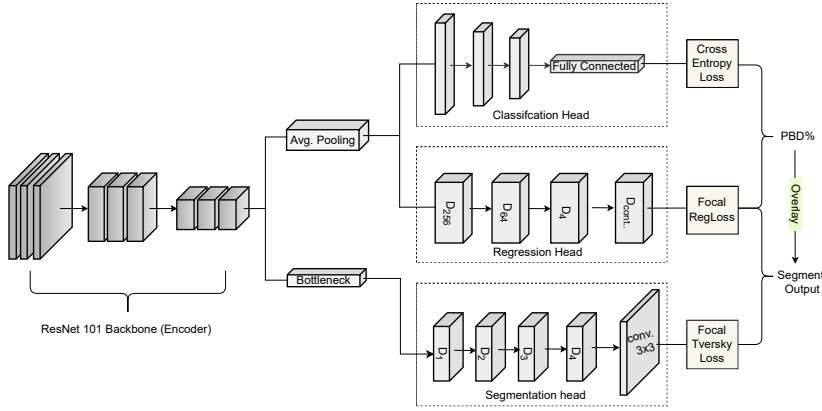

Figure 1: The deep MH-CNN architecture showing the process pipeline

### 2.1. Process Overview

Given an input mammogram $X$, the MH-CNN uses ResNet101 encoder to extract high-level features $F = f_{\text{encoder}}(X; \theta)$, where $\theta$ is encoder parameters. The shared feature map $F$ is then fed into three parallel heads: the segmentation head outputs $S = f_{\text{seg}}(F; \phi_{\text{seg}})$ and is optimized using the Focal Tversky loss, $\mathcal{L}_{\text{seg}} = \text{FocalTversky}(S, Y)$; where $Y$ is groundtruth mask. The regression head outputs $R = f_{\text{reg}}(F; \phi_{\text{reg}})$ with the FocalRegLoss, $\mathcal{L}_{\text{reg}} = \text{FocalRegLoss}(R, Y_{\text{reg}})$; and the classification head yields $C = f_{\text{cls}}(F; \phi_{\text{cls}})$ with the Cross Entropy Loss, $\mathcal{L}_{\text{cls}} = \text{CrossEntropy}(C, Y_{\text{cls}})$. Note that $\phi$ denotes the parameters of the respective heads. The total training loss, with weighting coefficients $\lambda$ is defined as,

$$\mathcal{L}_{\mathcal{T}} = \lambda_{\text{seg}}\mathcal{L}_{\text{seg}} + \lambda_{\text{reg}}\mathcal{L}_{\text{reg}} + \lambda_{\text{cls}}\mathcal{L}_{\text{cls}}$$

Using phase-based training we facilitate end-to-end learning for joint segmentation, regression, and classification tasks to achieve a unified model for streamlined PBD assessment.

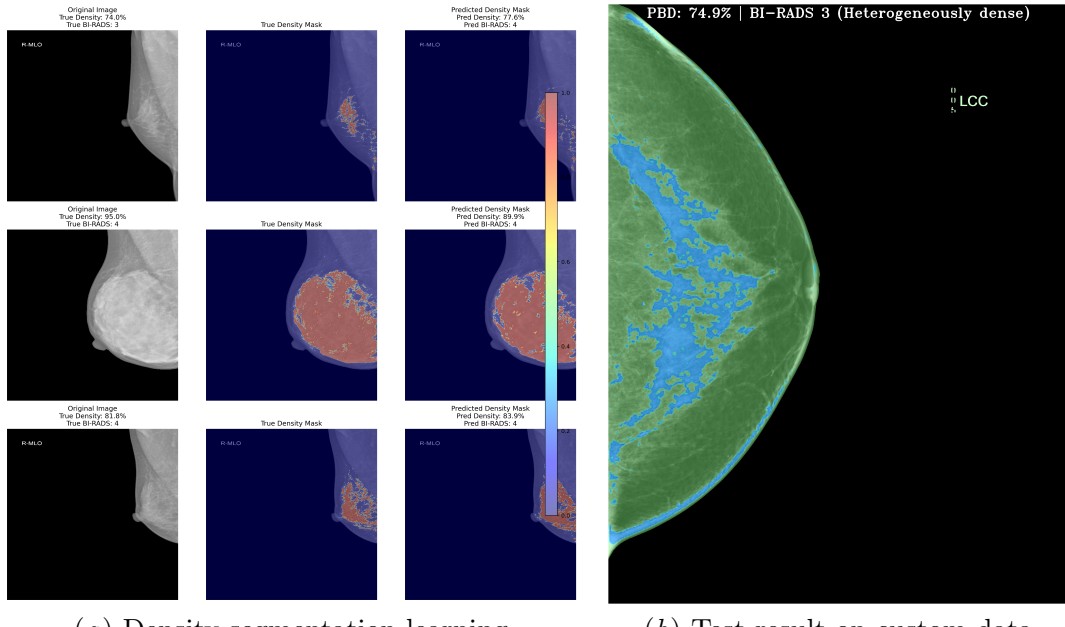

(*a*) Density segmentation learning    (*b*) Test result on custom data

Figure 2: Visualization results during (a) Training and (b) Inference

## 3. Experimental Settings and Outcomes

**Dataset:** The experiments were conducted on the VinDr-Mammo dataset (Nguyen et al., 2023) consisting of 20,000 mammograms obtained from 5,000 mammography exams (four images per patient). From this dataset, 590 mammo samples were randomly selected and split into training and validation subsets using an 80/20 ratio. Corresponding groundtruth density masks annotated by expert radiologists from (Gudhe et al., 2022) were used for model supervision.

**Training Setup:** The model was implemented using PyTorch and trained for 250 epochs with two Nvidia RTX 3080 GPUs. Training was prioritized in phases, initially focusing on segmentation, followed by regression, and finally BI-RADS classification. This phased strategy was adopted due to density imbalance within the skewed training data identified during preprocessing. Additional training configurations included the adoption of AdamW optimizer with a weight decay of 0.0001 and an initial learning rate of 5e-05.

**Result Summary:** Fig. 2(a) demonstrates accurate breast density segmentation, closely aligning with expert annotations. Fig. 2(b) showcases the model's robust generalization to unseen data through simultaneous density segmentation and PBD classification overlay. To our knowledge, this is the first unified approach for joint segmentation and classification of breast density without additional post-processing. Numeric results are listed in Table 1.

Table 1: Quantitative Results

| Metric | Outcome (%) |
|---|---|
| Dice Coefficient | ↑ 84.57 |
| MAE | ↓ 05.92 |
| Classification Accuracy | ↑ 80.51 |

## Acknowledgments

This work was supported by funding from the Chang Gung Memorial Hospital Research Project, under grant no. CLRPG3H0016

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
