# OpenReview forum: "Deep Multi-Head CNN for Unified Breast Density Segmentation and BI-RADS Estimation in 2D Mammograms"
_MIDL.io/2025/Short_Papers — MIDL 2025 - Short Papers_

### Official Review · Reviewer_Hy8n · 2025-04-29

**Rating:** 3
**Confidence:** 5

**Summary:**

The paper proposes a unified multi-head convolutional neural network (MH-CNN) for breast density segmentation, continuous density regression, and BI-RADS classification from 2D mammograms. The model is built on a ResNet101 backbone with three parallel heads corresponding to the different tasks, trained jointly in a phased approach. Experiments are conducted on a subset of the VinDr-Mammo dataset, and the model achieves a Dice coefficient of 84.57% for segmentation, a mean absolute error (MAE) of 5.92% for density regression, and an 80.51% accuracy for BI-RADS classification. The authors argue that this integrated approach could streamline clinical workflows by providing objective and automated assessments of breast density.

**Strengths:**

- The paper addresses a relevant clinical need by targeting both segmentation and classification of breast density in an integrated framework, which mirrors how radiologists evaluate mammograms.
- Combining these tasks into a single model is a practical idea that could reduce processing steps and deployment complexity. The use of a multi-head architecture leveraging a shared encoder is a reasonable design choice, allowing feature reuse across tasks, which can be beneficial especially when training data is relatively limited.
- The phased training approach to first stabilize segmentation, then regression, and then classification is thoughtful, considering the known imbalance in breast density distributions.
- The results achieved on the VinDr-Mammo dataset are promising, particularly the Dice score for segmentation, which is competitive with previous methods.
- The paper is mostly clear in describing the architecture and experimental setup, and the figures are helpful for understanding the workflow.

**Weaknesses:**

- Despite the practical relevance, the technical novelty of the proposed model is limited. Multi-head architectures and shared encoder strategies are common in medical imaging and beyond, and the model mainly combines established components without introducing significant innovations at the architectural or training level.
- The evaluation is relatively weak, and there is no external validation or cross-site testing, which raises concerns about the model’s robustness.
- There is little analysis of how well the tasks interact, for example, whether learning segmentation improves classification, or if task interference occurs.
- The choice of 80/20 split on such a small set without cross-validation is another limitation, given the inherent variability in breast density.
- No comparison with strong baselines or prior methods is included, making it difficult to assess how much the multi-task learning actually helps compared to single-task models.
- The metrics are reported but not analyzed statistically, so the strength of the results is hard to judge.

---

### Decision · Program_Chairs · 2025-05-01

Accept